# Application of the Maturity Model in Industrial Corporations

**Cihan Ünal [1],\*, Cemil Sungur [2] and Hakan Yildirim [3]**

[1] Department of Computer Programming, Hacettepe University, Ankara 06230, Turkey
[2] Faculty of Electrical Engineering, Konya Technical University, Konya 42250, Turkey; csungur@ktun.edu.tr
[3] Department of Computer Engineering, Beykent University, Istanbul 34500, Turkey; hakanyildirim@beykent.edu.tr
\* Correspondence: cihan.unal@hacettepe.edu.tr

**Abstract:** Enterprises need to evaluate for themselves whether they are ready for Industry 4.0 to survive and develop in the era of the Fourth Industrial Revolution. Therefore, it is necessary to conceptualize or develop an Industry 4.0 readiness and maturity model with basic model dimensions. The present study aimed to review the maturity models available in the literature and to develop and implement a comprehensive maturity model that would eliminate the problems in the existing models. Most maturity models developed lack vital dimensions such as laws, incentives, and corporate culture. While developing the model, AHP and expert opinions were used to determine the dimension weights. The model was applied to 87 businesses in various industries at the Ankara Chamber of Industry Industrial Park in Turkey. The developed model calculates the maturity level of the enterprise for six dimensions. The data on 61 corporations where Industry 4.0 technologies were adopted were analyzed based on demographic variables such as the year of establishment, industry, size, capital, and turnover. These findings demonstrated that Industry 4.0 was introduced recently in Turkey and businesses are required to take further steps to keep up with the global digital transformation. Since the number of industries and corporations that are aware of the Industry 4.0 technologies is limited in Ankara, Turkey, only a few businesses adopted the Industry 4.0 technologies. This developed model will make an important contribution to the literature with its unique dimensions. It would pave the way for further research in various industries in Turkey and other nations where Industry 4.0 investments are new.

**Keywords:** Industry 4.0; maturity model; readiness model; digitalization; supports





## 1. Introduction

Industry 4.0, Internet of things, big data, cloud computing, artificial intelligence, cyber-physical systems, sensors, 3D printers, autonomous robots, and emerging technologies such as cloud computing, work with each other over the network that connects people, machines, equipment, and interactive systems based on the exchange of data. Technology-based production, which has become a necessity with the increasing importance of Industry 4.0 technologies in our lives, provides businesses with a significant competitive advantage [1].

To survive in this new technological world, businesses have to prepare themselves for several changes. Businesses that cannot keep up with these changes are in danger of disappearing. The industry and the academy are constantly working to develop self-assessment models that can assess the readiness of enterprises for Industry 4.0. Achieving Industry 4.0 readiness has become a necessity for businesses today [2].

Enterprises need maturity models to determine Industry 4.0 levels, to determine their needs by self-evaluating, to compare themselves with their competitors, and to determine their plans and strategies correctly. In our study, a maturity model consisting of six dimensions is proposed to determine the Industry 4.0 maturity levels of companies [3].

After the invention of the steam engine in the 1700s, mechanical force replaced manual force, leading to the development of industry. The introduction of electricity in the 1900s

allowed mass production. In the second half of the 20th century, manufacturing was automated due to the invention of electronic circuits. In the 21st century, the introduction of the Internet led to prevalent cyber-physical systems [4]. These were the four significant cornerstones in the industry [5]. The current stage in industrial development is Industry 4.0. With the intensive use of information technologies, the transfer of business processes and information to electronic media quickly, saving money and labor, is called digital transformation. Digital transformation is not only associated with the industry, but it also affects societies, states, economies, education, etc. In the current study, only the industrial dimension of the concept of Industry 4.0 is discussed. The details of the correlation between industrial revolutions were discussed in a paper published by World Economic Forum Director Nicholas Davis (2016).

An important feature of the current age is awareness. Earlier, no one could predict the future and what would happen next, and no one needed to know.

The Fourth Industrial Revolution had a significant impact that increases every day. Klaus Schwab, President of the World Economic Forum, described the First Industrial Revolution as mechanical manufacturing, the Second Industrial Revolution as mass production, and the Third Industrial Revolution as the computer or digital revolution [6]. Germany's state strategy for Digital Transformation is called Industry 4.0. It was first announced at the Hannover Fair in 2011. After Germany, the United States of America announced the "America Produces Program" [7], China announced "Made in China 2025", Japan announced "Society 5.0" [8], and other nations similarly announced their digital transformation strategies. Turkey, on the other hand, announced the Turkish digital transformation program "Digital Turkey Roadmap" in June 2018 [9]. The "Turkish Industrial Revolution: Digital Turkey Roadmap" emphasized training the qualified workforce required for digital transformation. Furthermore, awareness and maturity levels were investigated in the "New Industrial Revolution Intelligent Production Systems Technology Roadmap" published by TUBITAK in 2016 [10]. Although the names and the domains of these programs were different in various countries, all nations aimed to determine strategies for the "New Industrial Revolution" and implement practical changes as a nation as soon as possible.

While scanning the literature, we focused on existing maturity models to develop a robust framework-based maturity model. Once the literature was examined, it was seen that there are almost no maturity models that form a bridge between academia and businesses. Moreover, existing maturity models in the literature were not fit into the dimensions of scope, relevance, completeness, and transparency. To fill this gap in the field, we define a maturity model and framework that covers all dimensions.

One of the most important observations of the field study is that an important obstacle stands in front of the manufacturing industry: the lack of trained labor force. In addition, the financial inadequacies of enterprises and the uncertainty of the return on investments related to Industry 4.0 are also very important barriers to the transition to Industry 4.0. The procedures that businesses need to adopt for "Industry 4.0" are included in roadmaps. However, to determine a comprehensive corporate roadmap, the readiness of the enterprise and possible obstacles to target achievements should be identified with diagnostic studies. Readiness and maturity could be determined by several expert consultants in different fields. In the current study, the SANOL Maturity Model developed for Turkish corporations was applied to 87 companies in different industries at Ankara Chamber of Commerce Industrial Park (ACCIP). A questionnaire was applied, and 26 participating businesses stated that they were not interested in Industry 4.0. A total of 61 businesses participated in the survey.

Based on the questionnaire responses, the overall score and six dimension scores were calculated for each enterprise, the model allowed to score the maturity of each corporation for transition to Industry 4.0, and the strengths and weaknesses of the enterprises were determined. Certain recommendations were presented to the businesses based on their scores.

The corporations could take the necessary steps after the application of the model based on their scores and readiness levels in each dimension and the recommendations associated with the maturity model.

In the current study, 61 participating businesses were classified and analyzed individually based on their years and fields of activity, capital, number of employees, and annual turnover.

This article is edited as follows: Part 2 consists of background and literature. The method is presented in Part 3. In Part 4, the findings obtained from the field study are given. In Part 5, the findings and the conceptual framework obtained are discussed; the general results of the article are also exhibited.

## 2. Background and Literature

Technologies associated with Industry 4.0 have radically changed the industries. Industry 4.0 has several advantages, such as a reduction in the time and cost of taking a product to the market.

### 2.1. Prevalent Technologies

Industry 4.0 could be briefly described as the digitization of production and services. Although the views vary, it is common knowledge that the concept was first introduced by the Communication Incentives Group of the Industry-Science Research Alliance at the 2011 Hannover Fair. The prominent technologies associated with Industry 4.0 are presented below [11]:

- Internet of things (including sensors).
- Autonomous robots.
- Simulation.
- Horizontal and vertical system integration.
- Big data and analytics.
- Cyber security.
- Additive manufacturing (3D).
- Augmented reality.
- Cloud computing.

#### 2.1.1. Internet of Things (IoT)

The Internet of Things is described as an extensive network of uniquely addressable objects and the communication of these objects on this network with a specific protocol [12]. The Internet of Things is a process where increasingly intelligent machines activate and deactivate themselves and other machines autonomously [13].

#### 2.1.2. Autonomous Robots

Manufacturers in various industries use robots in their operations. Robot technology has improved to become more competent, autonomous, flexible, and collaborative, and the cost of ownership of robots has become less expensive [14]. In the future, the interaction between the robots will increase, and they could work safely with humans with higher learning skills [15].

#### 2.1.3. Simulation

Simulation is described as the imitation of a technical and actual operation or a system in time. Thus, simulation is a systems model that includes predefined relationships between the objects in the system [16]. After the introduction of steam, electricity, and information systems to the factories and the business world during the first three industrial revolutions, simulation technologies are now introduced to the factories.

### 2.1.4. Vertical and Horizontal Systems Integration

Horizontal integration entails the integration of various representatives in business and collaboration models such as suppliers and customers. Vertical integration, on the other hand, includes smart operational systems such as in-house smart products, smart logistics networks, manufacturing, and marketing [17]. The approach improves logistics by integrating the corporate and customer data [18]. The economic outcomes of integration are the reduction of operating costs by shortening manufacturing time and improving product quality [19].

### 2.1.5. Cyber Security

Since digital transformation includes integrated systems fully equipped with digital technologies, secure communications and security are quite important. The main function of cyber security is to protect the data against the risks in cyberspace due to the complex ecosystem that includes the interaction between users, software, products, services, and businesses on Internet technologies and networks [20]. Although cyber security is one of the most important technologies that would facilitate digital transformation, unfortunately, it has not been emphasized adequately. It is usually remembered when there is a security vulnerability. Employees should be aware of their responsibilities to reduce cyber risks and have adequate skills and competencies [21].

### 2.1.6. Cloud Computing

Cloud computing is one of the pillars of Industry 4.0 [22]. Cloud computing is the general name for services to share data across computing devices. Thus, cloud computing is not a product, but a service. It allows data sharing across software at the source and the employment of the existing information services on an information network similar to power distribution [23].

### 2.1.7. Additive Manufacturing

Additive manufacturing is commonly known as three-dimensional (3D) printing [24]. It refers to the production of three-dimensional objects in layers based on virtual models [25].

The conventional manufacturing techniques can be categorized into three groups: cutting, carving, and molding [26]. These techniques require extensive time and costs. Three-dimensional printers significantly save time and costs. In digital transformation, 3D printers are expected to be used intensively due to their positive contributions to efficient inventory, eliminating the cost of mold production, and reducing manufacturing time.

### 2.1.8. Augmented Reality

Augmented reality entails direct or indirect observation of the real world where the content is augmented by computer-generated sounds, images, graphics, and GPS data. Augmented reality allows the user to observe digital parts placed on top of the physical parts [27]. Augmented reality briefly processes and augments reality digitally. The user could interact with available data through augmented reality technology. Artificial environmental data and elements could be compatible with the real world [28].

### 2.1.9. Big Data—Data Analytics

Big data is structured, or unstructured unusable data acquired over time. Big data describes huge databases that are digitized on the Internet and other networks. Big data includes technologies that support the processes of collecting and analyzing large data from various sources. These technologies are employed to process the data when the data size is not adequate for conventional storage, management, and analysis tools [29]. The most understandable and summarized form of the concept of "big data" is any dataset that does not fit in an Excel spreadsheet [30].

## 2.2. Theoretical Background

Enterprises need to evaluate whether they are ready for Industry 4.0 to survive during the Fourth Industrial Revolution [31]. Businesses need to attach importance to digitalization and technological transformation to survive in fierce market competition and increase efficiency, flexibility, and speed [32]. Industry 4.0 is a vision that defines the future of the industry [2]. Cooperation is important because Industry 4.0 is a technological revolution. Therefore, the support of national policies is important to make the productive system a reality. Structural difficulties can be seen by companies as obstacles to digital transformation, especially for small and medium-sized businesses that do not have the resources and tools [33]. Industry 4.0 is an advanced technology that can improve performance efficiency [34].

The SANOL Maturity Model was developed to measure the digital transformation maturity of enterprises. A healthy measurement is made using six different dimensions. AHP is used to determine the weight of the dimensions in the model.

The maturity index score of enterprises, the year of establishment of enterprises, sectors of enterprises, sizes of enterprises, organizational capital of enterprises, and how much it depends on the annual turnover of enterprises were studied in 87 enterprises. A total of 26 of the enterprises declared that they were not interested in digital transformation and did not participate in the study. A total of 61 businesses were surveyed. The results are given in the form of graphs and tables.

### Existing Readiness and Maturity Models

Maturity models are predetermined evolutionary stages to reach a target. These paths serve as a guide for the reorganization, restructuring, and renewal of existing skills to achieve excellence [35].

Businesses are exposed to competition due to market growth and are under pressure to advance parallel to constantly changing market conditions. Maturity models provide guidelines for the development of transformation competencies by initiating change. Maturity models serve as a guide and organizer for radical transformation, which requires digital evolution. The transformational effects of digitalization, and especially Industry 4.0, have become quite prevalent in recent years. Despite the increasing interest in Industry 4.0, guidance in Industry 4.0, perceptions, uncertainty about the related benefits and costs, and the level of the businesses in Industry 4.0 were not analyzed with an adequate model. Today, since almost all businesses employ Industry 4.0 standards, maturity models and Industry 4.0 maturity are quite important. The review of the related literature is shown in Table 1.

The review of the current maturity and readiness models revealed that the number of dimensions varied between 3 and 13. The limited number of dimensions prevented a robust analysis. However, the small number of dimensions does not pose a problem in certain sectors and special cases. The review demonstrated that although the dimension names were different, they served the same purpose in the analysis. For example, employees or individuals refer to the same dimension. Almost all models included the dimensions of technology, people, strategy, leadership, processes, and innovation under different names.

**Table 1.** Literature review.

| No | Model | Dimensions | The Difference Compared to the Developed SANOL MM |
|---|---|---|---|
| 1 | Industry 4.0 Maturity Model [36] | 3 dimensions (factory of the future, people and culture, strategy) | The number of dimensions is inadequate for a plausible analysis. |
| 2 | SME Maturity Model Assessment of IR4.0 Digital Transformation [37] | 7 dimensions (strategy and organization, smart factory, vertical and horizontal integration, distribution control, smart product, data-driven services, employees) | Does not include incentives and supports dimensions. |
| 3 | SSCM Assessment for Industry 4.0 [38] | 5 dimensions (management strategy and organization, collaboration, sustainable development, technology-based smart products, business-based smart operations) | Does not include incentives and supports dimensions. |
| 4 | Industry 4.0 Business Model [39] | 3 dimensions (value creation, value offer, value capture) | Inadequate number of dimensions. Does not include strategy, customers, employees, incentives, and supports dimensions. |
| 5 | Manufacturing Companies Industry 4.0 Adoption Model [40] | 3 dimensions (strategy, maturity, performance) | Inadequate number of dimensions for analysis. Does not include data and security, supports, and incentives dimensions. |
| 6 | ACATECH Industry 4.0 Maturity Index [41] | 4 dimensions (resources, information systems, organizational structure, culture) | Incentives and supports dimensions do not exist. |
| 7 | Enterprise 4.0 Assessment [42] | 7 dimensions (structure, design, management, culture, process, strategy, employee relationships) | Incentives and supports dimensions do not exist. |
| 8 | Industry 4.0 Maturity Model [43] | 5 dimensions (asset management, data governance, application management, process, transformation, organizational alignment) | Customers and suppliers, supports, and incentives dimensions do not exist. |
| 9 | Three Stages Maturity Model in SMEs towards Industry 4.0 [44] | 3 dimensions (envision, enable, enact) | Inadequate number of dimensions. Customers and suppliers, supports, and incentives dimensions do not exist. |
| 10 | Design Business Modelling for Industry 4.0 [45] | 7 dimensions (learning and growth perspective, competitiveness perspective, innovation perspective, operational & process level, financial level, strategic level, socio-environmental level) | Incentives and supports dimensions do not exist. |
| 11 | System Integration Maturity Model Industry 4.0 [46] | 4 dimensions (vertical integration, horizontal integration, cross-technology criteria, digital product development) | Lacks data and security, supports, and incentives dimensions. |
| 12 | Industry 4.0 Roadmap [47] | 13 dimensions (acceptance and application of new technology and media, professional competence, learning competence, corporate strategy, HR development strategy, organization and democratization, flexible working models, health and safety, information and communication, employer branding, change management, process orientation, knowledge management) | Lacks incentives and suppliers dimensions. |

**Table 1.** *Cont.*

| No | Model | Dimensions | The Difference Compared to the Developed SANOL MM |
|----|-------|------------|---------------------------------------------------|
| 13 | Industry 4.0 Maturity Model for Industry 4.0 Strategy [48] | 9 dimensions (strategy, leadership, customers, products, operations, culture, people, governance, technology) | Lacks incentives and suppliers dimensions. |
| 14 | Reference Architecture Model for the Industry 4.0 (RAMI4.0) [49] | 6 dimensions (business, functional, information, communication, integration, asset) | Lacks incentives and supports dimensions. |
| 15 | Design of an Assessment Industry 4.0 Maturity Model [50] | 6 dimensions (products and services, manufacturing, business model, strategy, supply chain, interoperability) | Lacks incentives and supports dimensions. |

When we examine the maturity models developed recently, the maturity model developed by Santos and colleagues consists of five dimensions. These dimensions are organizational strategy, structure, and culture; workforce; smart factories; smart processes; smart products and services. According to the respondents, the model is useful in establishing the initial diagnosis and forms a roadmap for continuing the practice [51].

Again, recently, a maturity model was developed by Rafael and his colleagues, particularly for SMEs, in the context of digitization and Industry 4.0 to diagnose the status of companies; it is an accessible, user-friendly, and comfortable self-assessment tool. Additionally, it allows its users to define a target value for each model size to decipher the gap between the current state and the desired state. The company can develop its future digitalization strategy by focusing on the areas that it considers the most critical, in comparison with the results achieved and the company's own set goals [52].

When a comprehensive analysis of all the dimensions in the current Industry 4.0 maturity model is performed, it is seen that the six most common dimensions used in these models are technology, people, strategy, leadership, process, and innovation, respectively. Therefore, it is expected that all or some of these six main dimensions will be used in the models to be developed. The fact that there is a dimension of corporate culture, laws, and incentives that are not in most models in the SANOL Maturity Model that we developed gives our model privilege.

The problems determined in the current models are indicated in Table 1. Almost all models do not include the supports and incentives dimensions. The support and tax exemptions for the import of Industry 4.0 technologies could positively or negatively affect the transition to Industry 4.0. Furthermore, the government incentives for the transition to Industry 4.0 could attract further businesses. Similarly, the removal of certain legal barriers is important for the transition of businesses to Industry 4.0. Thus, supports and incentives dimensions are important. The model developed in the current study includes supports and incentives dimensions. Thus, the SANOL Industry 4.0 Maturity Model eliminated a significant problem. It was observed that while certain models included a customer dimension, there was no supplier dimension. The supplier's dimension is crucial for horizontal integration. It is very important for communication between the firm and its suppliers and customers, especially in the manufacturing industry inventory. Certain models did not include a security dimension. Data security is vital for any business that adopts Industry 4.0 technologies. The SANOL Maturity Model that was developed based on the literature could be recommended for all manufacturing businesses. The SANOL Maturity Model could eliminate a key problem for manufacturing businesses and fill a significant gap in the literature. The developed model can also be used for different countries and different sectors.

## 3. Methodology

Manufacturing businesses need a reliable and valid roadmap to identify their readiness for a transition to Industry 4.0 and determine the steps required to advance and implement the process. To fulfill this requirement, the Industry 4.0 Maturity Model was developed. In the current study, the development, implementation, and analysis stages of the need-based model are discussed. The model targets manufacturing businesses. Certain businesses effectively use Industry 4.0 technologies, while others are insensitive. The current model would guide manufacturing businesses in their transition to Industry 4.0 standards or resolve their problems in the process.

The following steps were adopted in the development of the SANOL Industry 4.0 Maturity Model:

(1)　Determination of Industry 4.0 Maturity Model dimensions and sub-dimensions.
(2)　Determination of Industry 4.0 Maturity Model dimension and sub-dimension weights.
(3)　Determination of the analysis levels of the Industry 4.0 Maturity Model.
(4)　Determination of the Industry 4.0 maturity index.

The SANOL Industry 4.0 Maturity Model developed in the study aimed to expand the current models (Figure 1). Currently, almost all businesses have adopted Industry 4.0 technologies. Naturally, certain businesses are more predisposed to certain technologies, while others do not employ them at all. The technologies required by the type of business are employed. The employment of Industry 4.0 technologies varies between the industries. Certain businesses adopt additive manufacturing (3D) extensively, while others do not. This is valid for all Industry 4.0 technologies. The maturity model we developed was applied to manufacturing businesses and the maturity of the business was measured based on six Industry 4.0 dimensions. The analysis would reveal the weaknesses and strengths of the company to take the necessary steps towards Industry 4.0. The maturity model we developed has a corporate culture, laws, and incentives dimension that is not present in most models, which makes our model different. Therefore, the SANOL Maturity Model would fill a significant gap in the literature.

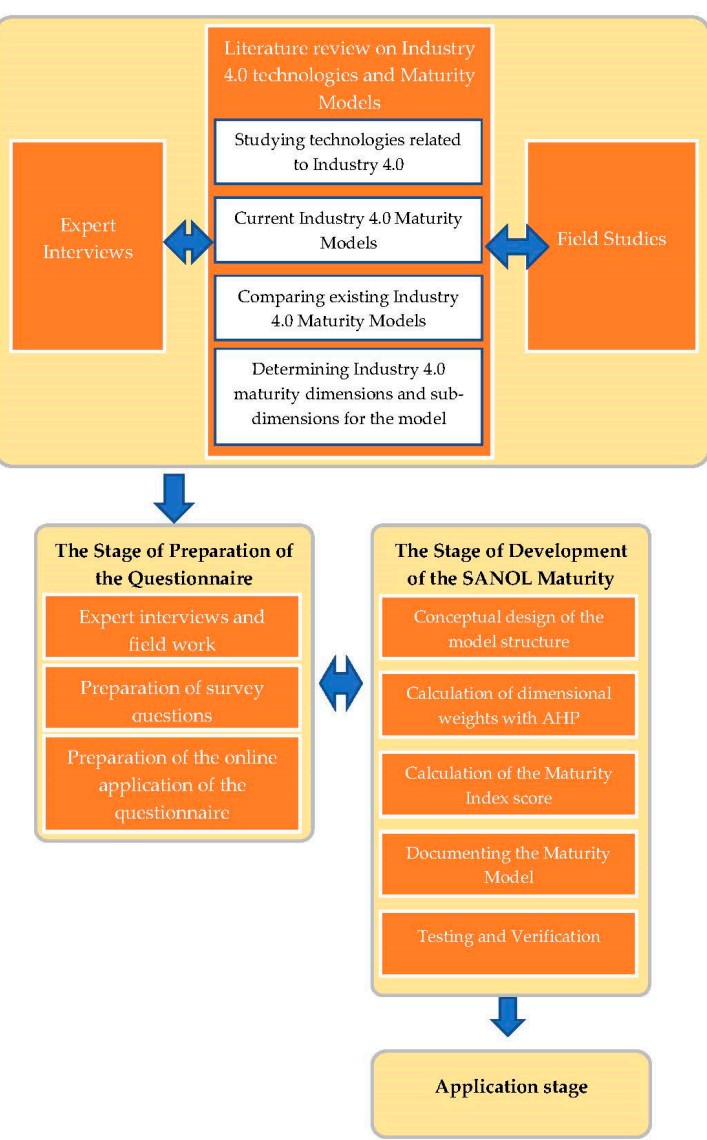

**Figure 1.** A multi-methodological research approach for the development of the SANOL Industry 4.0 Maturity Model (MM).

In the development of the proposed model, face-to-face interviews were conducted with 23 experts in Industry 4.0. During the interviews, the model was constantly updated based on the feedback, and the accuracy of the revisions was tested in the next interview.

### 3.1. The SANOL Industry 4.0 Maturity Model Dimensions and Sub-Dimensions

The SANOL Industry 4.0 Maturity Model developed for manufacturing enterprises was developed in meetings conducted with experts. SANOL was named after the Turkish words "sanayii" (industry) and "olgunluk" (maturity). The model dimensions and different weights of each dimension were also determined based on expert opinion. The experts included a chairman of the board, an IT manager, an R&D engineer, a digital transformation manager, a production manager, a human resources manager, and academicians. The dimensions were also determined based on the dimensions reported in maturity model studies in the literature. To test the accuracy of the dimensions, interviews were conducted with 38 individuals with expertise. The "supports and incentives" dimension was included based on the interviews. It was observed that businesses were generally reluctant about the adoption of new technologies. Thus, the incentives that would be adopted by regulatory and supervisory authorities (governments) would accelerate and facilitate the Industry 4.0 transition of the businesses. Thus, this dimension was considered necessary and included in the model.

The validity of this model was tested in fieldwork. The SANOL Maturity Model includes six dimensions and 22 sub-dimensions associated with these six dimensions:

Dimensions

Strategy and Management: The roadmap for the employment of Industry 4.0 technologies, implementation of these technologies in existing business models and goals. The support and willingness of the management for the transition.

Customers and Suppliers: Sharing data with customers and suppliers and customer and supplier competencies in Industry 4.0 technologies. Fast and reliable communication with customers and suppliers using horizontal and vertical integration.

Employees and Corporate Culture (Sense of Belonging): Employee competencies in Industry 4.0 technologies, the adoption of the institution, organizational values, and principles by the employees, and their willingness to take the company to a higher level. The employee sees the enterprise as his workplace and works according to it.

Technology (Products): The analysis of the adoption and employment of each Industry 4.0 technology.

Data and Security: The dimension includes the employment of data, data security, and cyber security in the business.

Supports and Incentives: The compliance of the business with Industry 4.0 supports and government incentives. In short, businesses benefit from the financial and moral support offered by the state

Dimensions and sub-dimensions are shown in Table 2.

### 3.2. The SANOL Maturity Model Survey

The survey questions were developed based on expert opinion and the interviews conducted with field specialists. The survey included 58 items. The number of questions related to each dimension is given in Table 3. In addition, 10 general questions were asked. The questionnaire consists of a total of 58 questions.

Participation in the survey was voluntary and the data were kept confidential.

In addition to general questions, the survey included items in six dimensions. General questions were not included in the analysis and were employed for classification purposes.

The survey included open-ended, yes or no, multiple-choice, multi-option, and 5-point Likert-type questions. A total of 11 general questions were not included in the analysis. The questions were determined based on field studies and tests conducted in several businesses to ensure the model validity. After the face-to-face interviews

were conducted with businesses, certain questions were added, and others were removed or revised. Furthermore, the questions were revised based on the opinion of the experts including academicians, a production manager, an IT manager, a project manager, and an R&D engineer. Based on the interviews conducted with these individuals, a questionnaire that included 58 questions was developed. Due to the COVID-19 pandemic, an online survey form was developed (the form can be accessed at https://form.jotform.com/210383307015040, accessed on 1 May 2022). The Turkish language questionnaire was applied to several businesses online. The English version of the questionnaire is available at https://www.jotform.com/form/213071375248959 (accessed on 1 May 2022).

**Table 2.** Dimensions and sub-dimensions.

| Dimension | Sub-Dimensions |
|---|---|
| Strategy and Management | Strategy |
| | Management |
| Customers and Suppliers | Customers |
| | Suppliers |
| Employees and Corporate Culture | Employees |
| | Corporate Culture |
| Technology | Big Data and Analytics |
| | Autonomous Robots |
| | Simulation |
| | Horizontal and Vertical Systems Integration |
| | Internet of Things (Including sensors) |
| | Cyber Security |
| | Additive Manufacturing (3D) |
| | Augmented Reality |
| | Cloud Computing |
| | Mobile Technologies |
| | Artificial Intelligence |
| | RFID and RTLS |
| Data and Security | Data |
| | Security |
| Supports and Incentives | Supports |
| | Incentives |

**Table 3.** Distribution of questions by dimensions.

| Dimension | Number of Questions |
|---|---|
| Strategy and Management | 11 |
| Customers and Suppliers | 3 |
| Employees and Corporate Culture | 5 |
| Technology (Products) | 12 |
| Data and Security | 7 |
| Supports and Incentives | 10 |

### 3.3. Testing the Validity of Dimension Weights with AHP (Analytical Hierarchy Process)

The SANOL Industry 4.0 Maturity Model we developed consists of strategy and management, customers and suppliers, employees and corporate culture, technology, data and security, legal supports, and benefiting from incentives. These dimensions are not of equal importance in all businesses. Various factors affect the dimension weights of the model. In some businesses, some dimensions are very important, while others are less important. For example, data and security may be very important, while customers and suppliers may be less important. Since the model was developed for enterprises in the industry, this was taken into account when determining the weight of the dimensions. Likewise, the weight of the sub-dimensions may differ. While determining the weight of the dimensions, expert opinions were taken and the analytical hierarchy process (AHP) was used as a decision support tool. The analytical hierarchy process (AHP) technique, developed by Thomas L. Saaty in the 1970s, is a structured method for organizing and analyzing complex decisions based on mathematics and psychology. The method frequently used in production management problems is a flexible tool that can be applied to the hierarchy and used to support decision-making when more than one criterion is involved. In the AHP method, the weights of the dimensions are determined based on pairwise comparisons. Pairwise comparisons at different levels of hierarchy are used to estimate the weights of dimensions and sub-dimensions, resulting in the decision matrix. Mathematical methods are used to find the eigenvector of the decision matrix and then the consistency ratio is looked at to evaluate the consistency of the decisions. The application steps of the AHP method for our model are as follows.

Step 1: Defining the problem and forming the hierarchical structure.

In the hierarchical structure, dimensions and sub-dimensions must be determined correctly. In addition, since the correct determination of the number of factors that will affect the result and the detailed definition of each factor is important in terms of making pairwise comparisons consistent and logical, the factors affecting the decision points should be determined correctly.

Step 2: Making pairwise comparisons between each dimension and other dimensions.

The inter-factor comparison matrix is an n x n square matrix (Table 4). The matrix components on the diagonal of this matrix take the value 1. While creating the pairwise comparison matrix, the importance scale, which determines the importance of the dimensions compared to the other dimensions, is used. The superiority of each dimension over the other dimension is indicated numerically. Since the number of dimensions is 6, a $6 \times 6$ matrix was used. Since each dimension is considered to have a superiority of 1, the values on the diagonal of the matrix are 1. Strategy and management are 3 times more superior to customers and suppliers but are half as important to the technology dimension. The strategy and management dimension has the same importance as the data and security dimension. The matrix is filled in this way for all dimensions, and column totals are taken.

Step 3: Calculating the importance of each item related to other items.

After the pairwise comparison matrices are created, there is a relative weight vector showing the importance of each item in the relevant matrix relative to other items. For each dimension, there is the weight of the other dimensions. Dimension weight values are found by dividing each element of the matrix by the column total. The weight of the dimension is found by taking the sum of the rows. Dimension weights are given out of 6. Therefore, the sum of the weights of all dimensions is 6 (Tables 5 and 6).

Step 4: Performing the consistency test, which reveals whether the severity levels are calculated correctly or not.

The AHP method proposes a process for measuring consistency in pairwise comparisons. It provides the opportunity to test the consistency of the priority vector found with the resulting consistency ratio (CR), and, therefore, the consistency of one-to-one comparisons between factors. For each dimension, row totals are taken and divided by their weights. The value for each dimension is summed and divided by the number of dimensions. It is found that $\lambda$max = 6.121 (Table 7).

**Table 4.** Factor comparison matrix.

| DIMENSIONS | Strategy and Management | Customers and Suppliers | Employees and Corporate Culture | Technology | Data and Security | Supports and Incentives |
|---|---|---|---|---|---|---|
| Strategy and Management | 1 | 3 | 3 | 0.500 | 1 | 5 |
| Customers and Suppliers | 0.333 | 1 | 0.500 | 0.200 | 0.250 | 1 |
| Employees and Corporate Culture | 0.333 | 2 | 1 | 0.200 | 0.200 | 1 |
| Technology | 2 | 5 | 5 | 1 | 1 | 5 |
| Data and Security | 1 | 4 | 5 | 1 | 1 | 5 |
| Supports and Incentives | 0.200 | 1 | 1 | 0.200 | 0.200 | 1 |
| TOTAL | 4.867 | 16.000 | 15.500 | 3.100 | 3.650 | 18.000 |

**Table 5.** Weights of dimensions.

| DIMENSIONS | Strategy and Management | Customers and Suppliers | Employees and Corporate Culture | Technology | Data and Security | Supports and Incentives | DIMENSIONS WEIGHT |
|---|---|---|---|---|---|---|---|
| Strategy and Management | 0.205 | 0.188 | 0.194 | 0.161 | 0.274 | 0.278 | 1.300 |
| Customers and Suppliers | 0.068 | 0.063 | 0.032 | 0.065 | 0.068 | 0.056 | 0.352 |
| Employees and Corporate Culture | 0.068 | 0.125 | 0.065 | 0.065 | 0.055 | 0.056 | 0.433 |
| Technology | 0.411 | 0.313 | 0.323 | 0.323 | 0.274 | 0.278 | 1.920 |
| Data and Security | 0.205 | 0.250 | 0.323 | 0.323 | 0.274 | 0.278 | 1.652 |
| Supports and Incentives | 0.041 | 0.063 | 0.065 | 0.065 | 0.055 | 0.056 | 0.343 |

**Table 6.** Weights of dimensions (out of 6).

| DIMENSIONS | Strategy and Management | Customers and Suppliers | Employees and Corporate Culture | Technology | Data and Security | Supports and Incentives | TOTAL |
|---|---|---|---|---|---|---|---|
| Strategy and Management | 1.300 | 1.055 | 1.299 | 0.960 | 1.652 | 1.715 | 7.981 |
| Customers and Suppliers | 0.433 | 0.352 | 0.216 | 0.384 | 0.413 | 0.343 | 2.142 |
| Employees and Corporate Culture | 0.433 | 0.704 | 0.433 | 0.384 | 0.330 | 0.343 | 2.627 |
| Technology | 2.599 | 1.759 | 2.164 | 1.920 | 1.652 | 1.715 | 11.810 |
| Data and Security | 1.300 | 1.407 | 2.164 | 1.920 | 1.652 | 1.715 | 10.159 |
| Supports and Incentives | 0.260 | 0.352 | 0.433 | 0.384 | 0.330 | 0.343 | 2.102 |

**Table 7.** Weight sum of dimensions.

| Dimensions | Row Total | Dimension Weight | Row Total/Dimension Weight |
|---|---|---|---|
| Strategy and Management | 7.981 | 1.300 | 6.141 |
| Customers and Suppliers | 2.142 | 0.352 | 6.087 |
| Employees and Corporate Culture | 2.627 | 0.433 | 6.069 |
| Technology | 11.810 | 1.920 | 6.150 |
| Data and Security | 10.159 | 1.652 | 6.148 |
| Supports and Incentives | 2.102 | 0.343 | 6.129 |

Average = 6.121; $\lambda$max = 6.121.

n: Number of criteria. Since we have 6 dimensions, n = 6 is taken.
Consistency index (CI):

$$CI = \frac{\lambda max - n}{n - 1}$$

$$CI = \frac{6.121 - 6}{5}$$

$$CR = \frac{CI}{RI}$$

Since the number of dimensions is 6, the RI is taken as 1.252 from the table.

**Random consistency index (RI).**

| n | 1 | 2 | 3 | 4 | 5 | 6 | 7 | 8 | 9 | 10 |
|---|---|---|---|---|---|---|---|---|---|---|
| RI | 0 | 0 | 0.525 | 0.882 | 1.115 | 1.252 | 1.341 | 1.404 | 1.452 | 1.484 |

$R = \frac{0.024}{1.252} = 0.019$

The CR (consistency ratio) is 0.019. If CR was greater than 0.10, we would not be able to make model predictions with these weights. Since it is less than 0.10, the dimension weights in our model are valid. The validity of the dimension weights of the SANOL Industry 4.0 Maturity Model that we developed was tested. To show the dimension weights in the 6th system as a percentage, the weight of the dimension is divided by 6 and multiplied by 100. It is shown in Table 8. Percentage dimensional weights with defects were rounded off as follows. Excel was used for AHP calculations.

**Table 8.** Dimension weight percentages.

| Dimensions | Dimension Weight | Dimension Weight in Percent (Fractional) | Dimension Weight in Percent (Rounded) |
|---|---|---|---|
| Strategy and Management | 1.300 | 21.659 | 22 |
| Customers and Suppliers | 0.352 | 5.864 | 6 |
| Employees and Corporate Culture | 0.433 | 7.215 | 7 |
| Technology | 1.920 | 32.006 | 32 |
| Data and Security | 1.652 | 27.540 | 28 |
| Supports and Incentives | 0.343 | 5.716 | 5 |
| **Total** | **6** | **100** | **100** |

### 3.4. The SANOL Maturity Model Dimension Weights

The dimensions are not of equal importance for each business. Experts were consulted to determine the weight of the dimensions and sub-dimensions. The estimated size weights were determined with the arithmetic average of the expert suggestions. The validity of the dimension weight was then tested with the analytical hierarchy process (AHP). Based on the findings, a pairwise comparison matrix was produced. Since the consistency ratio (CR) was adequate, the weight of the dimensions was calculated as percentages and presented in Table 9.

**Table 9.** Dimension weights.

| Dimension | Weight (Percentage) |
|---|---|
| Strategy and Management | 22 |
| Customers and Suppliers | 6 |
| Employees and Corporate Culture | 7 |
| Technology (Products) | 32 |
| Data and Security | 28 |
| Supports and Incentives | 5 |

As seen in Table 9 and Figure 2, the most significant dimension was "technology", followed by the "data and security", "strategy and management", "employees and corporate culture", "customers and suppliers", and "supports and incentives" dimensions. It was observed that the last three dimension weights were similar.

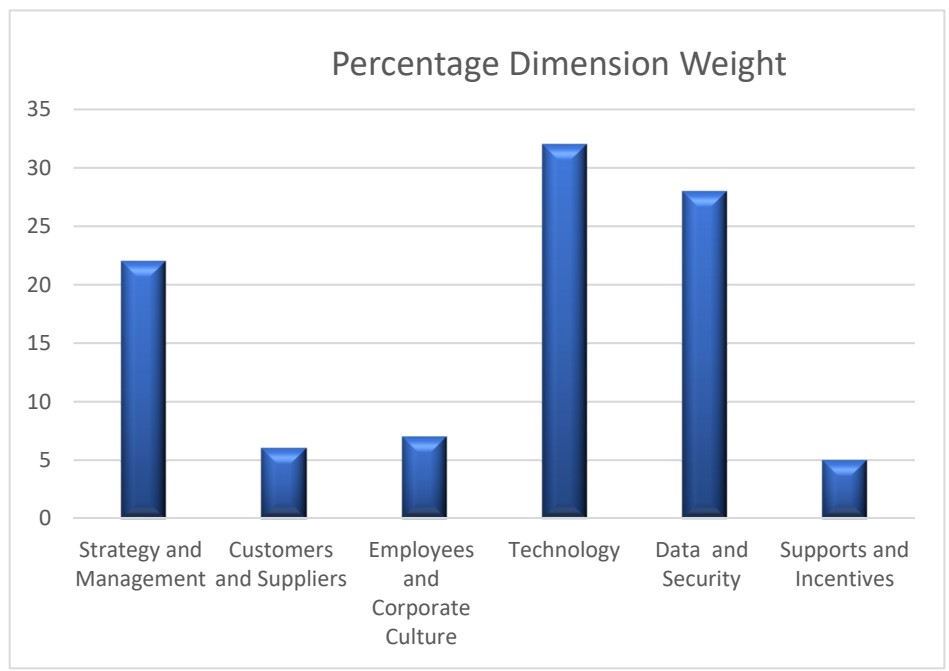

**Figure 2.** Percentage dimension weight.

### 3.5. The SANOL Maturity Model Analysis Levels

The literature review revealed that the current models had at least four, and at most six, analysis levels. Expert opinion was also obtained to determine the SANOL Industry 4.0 Maturity Model analysis levels. For the current model, six analysis levels were determined. The index score varied between 0 and 5. This range was divided into sox analysis levels (0: uninitiated, 1: initiated, 2: intermediate, 3: advanced, 4: very advanced, and 5: mature). The SANOL Industry 4.0 Maturity Model analysis levels are shown in Table 10.

**Table 10.** IS: index scores.

| Maturity Level | Detail | Index Score Weight |
|---|---|---|
| *Level 0* | *Uninitiated* | *IS = 0* |
| A SANOL Industry 4.0 Maturity Model Index Score of 0 (zero) indicates that the company is not interested in Industry 4.0 and has not taken any transitional steps. | | |
| *Level 1* | *Initiated* | *0 < IS ≤ 1* |
| A SANOL Industry 4.0 Maturity Model Index Score between zero and one indicates that the company has recently initiated work towards Industry 4.0 or employs a limited number of Industry 4.0 technologies. Management support is limited, and employees have little knowledge of Industry 4.0 technologies. The company has adopted almost no technology to communicate with customers and suppliers. There is no provision for data security. The company has no or little knowledge of related supports and incentives. | | |
| *Level 2* | *Intermediate* | *1 < IS ≤ 2* |
| The company has adopted the related technologies at a moderate level. The business has adopted an Industry 4.0 strategy. The management believes in the advantages of the transition to Industry 4.0. Employees are also moderately predisposed to Industry 4.0 technologies and benefits. Industry 4.0 technologies are employed in communications with customers and suppliers. Data security is emphasized. The company has moderate knowledge of supports and incentives. | | |
| *Level 3* | *Advanced* | *2 < IS ≤ 3* |
| The company employs Industry 4.0 technologies well. The management and employees recognize the advantages of Industry 4.0 technologies. Management supports the transition. The skills of the employees are satisfactory. Customers and suppliers have also adopted Industry 4.0 technologies. Data security is high and the business complies with the supports and enjoys available incentives. | | |
| *Level 4* | *Very advanced* | *3 < IS ≤ 4* |
| The company employs Industry 4.0 technologies at a very advanced level. However, the company is not fully mature. Management supports Industry 4.0 technologies. Management and employees have adopted Industry 4.0 and their competencies are at a very good level. Data security levels are very good. The company complies with the supports and utilizes the available incentives at an advanced level. | | |
| *Level 5* | *Mature* | *4 < IS ≤ 5* |
| The company employs Industry 4.0 technologies at a very good and mature level. Industry 4.0 has been adopted by the management and the management provides support for Industry 4.0 technologies. Employees are competent in Industry 4.0 technologies, Industry 4.0 technologies are employed in all departments, and by the customers and suppliers at an advanced level. Data security is prioritized at all levels. The company fully complies with related supports benefit from available incentives. The company has fully completed the transition to Industry 4.0. The level is the objective of companies in the transition to Industry 4.0. | | |

### 3.6. Calculation of the SANOL Maturity Index

During the development of the SANOL Maturity Index, 47 items in 6 dimensions were used. Eleven out of 58 survey questions were not used in the analysis, since these questions were catalog questions, such as e-mail address and year of establishment, that would not affect the analysis but could be used in classification. Initially, each of the dimension and sub-dimension scores were calculated separately. When a sub-dimension included more than one question, the mean score for these questions was calculated to determine the sub-dimension score. When there was only one question in a sub-dimension, the question score was accepted as the sub-dimension score. The dimension score was calculated by adding the weighed sub-dimension scores. The dimension weights are shown in Table 4. The Industry 4.0 Maturity Index (Supplementary Materials) was calculated by adding all weighed dimension scores.

E: Industry 4.0 maturity index score (between 1 and 5).
B: Dimension maturity score (between 1 and 5).
A: Sub-dimension maturity score.
P: Question score.
k: Number of questions in a sub-dimension.
n: Number of dimensions.
M: Number of sub-dimensions in a dimension.
t: Overall sub-dimension weight.
w: Sub-dimension weight in a dimension.

g: Overall dimension weight.

$Ai = \sum_{j=1}^{k} (Pj)/k$

$Bi = \sum_{i=1}^{m} Ai * wI$

SANOL Maturity Index score could be calculated with two methods [1].

Based on sub-dimensions:

$E = \sum_{i=1}^{n} \sum_{j=1}^{m} Aij * tij$

Based on dimensions:

$E = \sum_{i=1}^{n} (Bi * gi)$

## 4. Implementation of the SANOL Maturity Model

The model was applied to 61 corporations in different industries, and the findings are shown in Table 11. The highest scale score was 5. According to the year of establishment, the number of businesses established in the last 10 years is 20, the number of businesses established between the last 10 and 19 years is 14, and the number of businesses operating for more than 20 years is 27. The number of enterprises by sector is 8 machines, 10 services, 13 electronics, 13 manufacturing, 9 logistics, and 8 others, respectively. The number of enterprises with fewer than 250 employees is 44, and the number of enterprises with more than 250 is 17 units. The number of enterprises established with domestic capital is 49, the number of enterprises with foreign capital is 7, and the number of enterprises with both domestic and foreign capital is 5. The number of enterprises with an annual turnover of less than 100 million Turkish liras is 20, while the number of enterprises with a turnover of more than 100 million Turkish liras is 41.

**Table 11.** Mean company scores in all dimensions.

| Dimension Scores | |
|---|---|
| Strategy and Management | 3.39 |
| Customers and Suppliers | 3.30 |
| Employees and Corporate Culture | 3.45 |
| Technology (Products) | 3.44 |
| Data and Security | 3.40 |
| Supports and Incentives | 4.02 |

As it consists of enterprises that have started and are about to start the digital transformation from enterprises located in the Organized Zone of the Ankara Chamber of Industry, the survey dimension scores turned out to be close to each other. A total of 26 businesses that are not interested in digital transformation did not participate in the survey. The maturity index scores of the surveyed enterprises are also very close to each other. It is estimated that very different maturity index scores will be obtained by conducting surveys in different industrial regions and different countries. The survey can be applied to different regions, different sectors, and different countries to contribute to the development of the model (Figure 3).

The maturity index score is calculated by adding weighted dimension scores:

$$E = \sum_{i=0}^{n} Bi * gi \tag{1}$$

Maturity index score =
3.39 * 0.22 + 3.30 * 0.06 + 3.45 * 0.07 + 3.44 * 0.32 + 3.40 * 0.28 + 4.02 * 0.05).
Index score = 3.40.

The analysis of the corporations based on the years of activity revealed that the dimension scores of the old enterprises were higher when compared to newer businesses. The mean score of 20 years or newer businesses was 4.00, while the mean score of the businesses established within 9 years was 2.73 (Table 12).

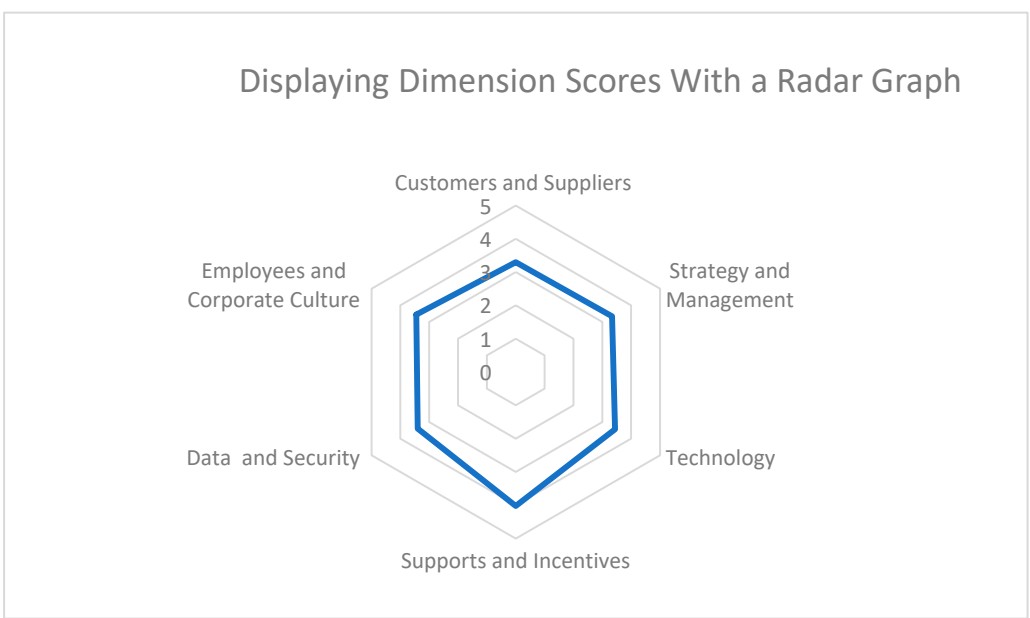

**Figure 3.** Dimension scores.

The analysis of the corporations based on industry revealed that the services industry corporations scored the lowest. The index scores for logistics, manufacturing, and other industries were above average (Table 13).

The analysis of the companies based on founding capital revealed that domestic companies had the highest index and dimension scores, followed by the companies with foreign capital. The mixed-capital businesses scored the lowest mean points. It was observed that the employment of Industry 4.0 technologies by foreign businesses was lower since they were predominantly in the services industry (Table 14).

The analysis of the companies based on the number of employees, namely, the company size, revealed that the mean score of the businesses with 251–1000 employees was the highest, followed by enterprises with 1000+ employees. The companies with 1–9 employees scored highest in the customers and suppliers dimension, while the businesses with 1000+ employees scored highest in the technology and supports and incentives dimensions. Larger enterprises scored higher (Table 15).

Businesses with an annual turnover of TRY 100–250 million scored highest in the strategy and management, customers and suppliers, and employees and corporate culture dimensions. Businesses with an annual turnover of TRY 250–500 million scored highest in the technology, data, security, supports, and incentives dimensions. The overall score of the businesses with an annual turnover of TRY 250–500 million was highest, followed by the businesses with an annual turnover of TRY 100–250 million. Enterprises with high annual turnover had higher scores (Table 16).

The most widely used Industry 4.0 technologies in enterprises are autonomous robots, mobile technologies, the Internet of Things, artificial intelligence, cloud computing, and additive manufacturing, respectively.

Autonomous robots and additive manufacturing are very widely used in the manufacturing sector, while the Internet of Things and artificial intelligence come to the fore in the logistics sector.

Autonomous robots, simulation, and cloud computing are the most widely used technologies related to Industry 4.0 in the machinery and electronics sector.

**Table 12.** Active years in business.

| Active Years | N | Percentage | Index Score | Strategy and Management | Customers and Suppliers | Employees and Corporate Culture | Technology | Data and Security | Supports and Incentives |
|---|---|---|---|---|---|---|---|---|---|
| Less than 10 | 20 | 32.79 | 2.73 | 2.45 | 2.55 | 2.82 | 2.84 | 2.71 | 3.45 |
| 10–19 | 14 | 22.95 | 3.35 | 3.53 | 3.03 | 3.13 | 3.27 | 3.33 | 3.96 |
| More than 20 | 27 | 44.26 | 4.00 | 4.01 | 3.99 | 4.08 | 3.96 | 3.94 | 4.48 |
| Total/AVG | 61 | 100.0 | 3.44 | 3.39 | 3.30 | 3.45 | 3.44 | 3.40 | 4.02 |

**Table 13.** The industries of the participating companies.

| Industry | N | Percentage | Index Score | Strategy and Management | Customers and Suppliers | Employees and Corporate Culture | Technology | Data and Security | Supports and Incentives |
|---|---|---|---|---|---|---|---|---|---|
| Machinery | 8 | 13.12 | 3.40 | 3.45 | 2.70 | 3.36 | 3.41 | 3.46 | 3.64 |
| Services | 10 | 16.39 | 3.10 | 2.94 | 3.38 | 3.24 | 3.10 | 2.97 | 3.95 |
| Electronic | 13 | 21.31 | 3.34 | 3.12 | 2.94 | 3.27 | 3.46 | 3.37 | 4.03 |
| Manufacturing | 13 | 21.31 | 3.58 | 3.53 | 3.54 | 3.76 | 3.54 | 3.52 | 4.17 |
| Logistics | 9 | 14.75 | 3.58 | 3.64 | 3.58 | 3.43 | 3.52 | 3.54 | 4.19 |
| Other | 8 | 13.12 | 3.66 | 3.80 | 3.66 | 3.62 | 3.59 | 3.56 | 4.07 |
| Total/AVG | 61 | 100.0 | 3.44 | 3.39 | 3.30 | 3.45 | 3.44 | 3.40 | 4.02 |

**Table 14.** The capital of the participating companies.

| Capital | N | Percentage | Index Score | Strategy and Management | Customers and Suppliers | Employees and Corporate Culture | Technology | Data and Security | Supports and Incentives |
|---|---|---|---|---|---|---|---|---|---|
| Domestic | 49 | 80.33 | 3.50 | 3.44 | 3.40 | 3.52 | 3.51 | 3.44 | 4.05 |
| Foreign | 7 | 11.47 | 3.21 | 3.31 | 3.07 | 3.22 | 2.95 | 3.27 | 4.25 |
| Mixed | 5 | 8.20 | 3.16 | 2.97 | 2.57 | 3.08 | 3.38 | 3.14 | 3.42 |
| Total/AVG | 61 | 100.0 | 3.44 | 3.39 | 3.30 | 3.45 | 3.44 | 3.40 | 4.02 |

**Table 15.** Number of employees in participating companies.

| # of Employees | N | Percentage | Index Score | Strategy and Management | Customers and Suppliers | Employees and Corporate Culture | Technology | Data and Security | Supports and Incentives |
|---|---|---|---|---|---|---|---|---|---|
| 1–9 | 7 | 11.48 | 3.38 | 3.16 | 3.53 | 3.45 | 3.40 | 3.41 | 3.85 |
| 10–50 | 13 | 21.31 | 3.31 | 3.27 | 3.23 | 3.32 | 3.28 | 3.32 | 3.82 |
| 51–250 | 24 | 39.34 | 3.39 | 3.40 | 3.23 | 3.38 | 3.41 | 3.25 | 4.13 |
| 251–1000 | 11 | 18.03 | 3.69 | 3.69 | 3.36 | 3.81 | 3.58 | 3.83 | 3.89 |
| 1000+ | 6 | 9.84 | 3.49 | 3.30 | 3.32 | 3.36 | 3.66 | 3.35 | 4.50 |
| Total/AVG | 61 | 100.0 | 3.44 | 3.39 | 3.30 | 3.45 | 3.44 | 3.40 | 4.02 |

**Table 16.** Annual turnover of the participating companies.

| Turnover | N | Percentage | Index Score | Strategy and Management | Customers and Suppliers | Employees and Corporate Culture | Technology | Data and Security | Supports and Incentives |
|---|---|---|---|---|---|---|---|---|---|
| TL < 1 million | 2 | 3.28 | 2.22 | 1.95 | 2.51 | 3.02 | 2.31 | 1.92 | 3.02 |
| TL 1–10 million | 3 | 4.92 | 3.49 | 3.08 | 3.50 | 3.29 | 3.65 | 3.66 | 3.49 |
| TL 10–50 million | 8 | 13.11 | 3.40 | 3.29 | 3.30 | 3.30 | 3.32 | 3.54 | 3.40 |
| TL 50–100 million | 7 | 11.48 | 3.11 | 3.16 | 3.18 | 3.32 | 3.08 | 2.90 | 3.11 |
| TL 100–250 million | 13 | 21.31 | 3.57 | 3.56 | 3.65 | 3.56 | 3.57 | 3.47 | 4.14 |
| TL 250–500 million | 15 | 24.59 | 3.64 | 3.51 | 3.45 | 3.52 | 3.73 | 3.58 | 4.27 |
| TL > 500 million | 13 | 21.31 | 3.44 | 3.54 | 2.91 | 3.52 | 3.35 | 3.45 | 4.07 |
| Total/AVG | 61 | 100.0 | 3.44 | 3.39 | 3.30 | 3.45 | 3.44 | 3.40 | 4.02 |

## 5. Conclusions and Discussion

The SANOL maturity model dimensions and sub-dimensions were determined based on expert opinion, and AHP was employed to identify the dimension weights. To determine the model validity, face-to-face interviews were conducted with several businesses, and the items were constantly revised. The survey was administered to a large number of businesses both face-to-face and online. The SANOL Maturity Model was different from previous models since it included the supports and incentives dimension in the analysis. Most businesses were reluctant to transition to Industry 4.0 technologies. Financial and moral government incentives would reduce the reluctance of the businesses to transition to Industry 4.0. Certain state decrees are also effective in the transition to Industry 4.0. For example, certain factors, such as the taxation of robots, would negatively affect the transition to Industry 4.0. Thus, the supports and incentives dimension is an important factor in the analysis, differentiating the SANOL maturity model.

The questions employed in the analysis were more comprehensive when compared to the existing models. The survey could be applied both face-to-face and online. Due to the COVID-19 pandemic, online surveys allowed the authors to reach more businesses. Furthermore, businesses could easily learn their maturity levels and maturity index scores online. Rapid advances in informatics prioritized the adoption of the innovations by the companies.

In the current study, the SANOL maturity model was applied to 61 businesses in various industries at Ankara Chamber of Industry Industrial Park. The findings were analyzed based on the six-dimensional SANOL model and the results were discussed. The AHP method was employed to determine the dimension weights, and the survey was completed by the employees in the accounting, human resources, and IT departments.

The survey was analyzed based on five demographic properties.

The analysis of the corporations based on the years of activity revealed that the dimension scores of older businesses were higher when compared to newer businesses.

The analysis of the corporations based on industry revealed that the businesses in other, manufacturing, and logistics industries scored highest.

The analysis of the corporations based on founding capital demonstrated that domestic enterprises had the highest overall and dimension scores, followed by the companies with foreign capital. The companies with the lowest mean score were the domestic and foreign partnerships. It was observed that the employment of Industry 4.0 technologies by foreign businesses was lower since they were predominantly in the services industry.

The dimension scores were determined for the supports and incentives for all industries. The highest scores were observed in this dimension.

The literature review focused on the current maturity models to develop a viable maturity model based on a strong framework. In previous chapters, extensive discussions were conducted, taking into account the benefits of Industry 4.0 and the potential difficulties that businesses will face when implementing these technologies.

When the literature is examined, it is seen that maturity model studies that provide a bridge between academia and enterprises are lacking. None of the maturity models examined in the literature fully cover the full dimensions of scope, fit for purpose, completeness, and openness. To bridge the gap in the literature, we defined the creation of a maturity model and framework covering all dimensions.

According to our findings, it seems that the most important obstacle to the manufacturing industry is the lack of a trained labor force. In addition, the financial inadequacies of enterprises and the uncertainty of the return on investments made are also important obstacles to the transition to Industry 4.0. The literature review revealed that only a few maturity models were developed in Turkey, such as those published by Akdil [47] and Gökalp [42]. Basl took advantage of the past literature to highlight some dimensions from some existing Industry 4.0 models but did not identify the most common ones, through a systematic review [53]. In terms of the variety of sizes of Industry 4.0 readiness models, Brozzi classified that some models are wide and have many sizes for measuring Industry 4.0

readiness, while others are narrow with a small number of evaluation sizes [54]. The need for more and various dimensions in the evaluation makes it difficult for the organizational system to converge with Industry 4.0 technologies [55]. The level of maturity is influenced by the stage of maturity of competing companies. In other words, if other enterprises in the same market are included in the survey, the maturity stage of the market is assessed; otherwise, they will not be taken into account [43].

The SANOL maturity model, which was also developed in Turkey, attempts to fill a significant gap. The model would allow the businesses to determine their maturity levels and take the necessary steps when necessary. The developed SANOL model could also inspire new models.

Reasons such as poor returns on investment are among the most significant obstacles to the transition to Industry 4.0.

Industry 4.0, which is considered to be the beginning of the Fourth Industrial Revolution, reveals the fact that many companies are developing and changing their existing business processes and business models in the direction of Industry 4.0. Industry 4.0 maturity models can contribute to achieving competitive advantage by starting the process of change in enterprises.

Since the phenomenon of digital transformation in manufacturing enterprises is still at the stage of emergence, there is no generally accepted standard maturity model for assessing the readiness of enterprises for Industry 4.0. It was determined that a new model was proposed by each researcher in the applied studies. These proposed models were developed mainly to evaluate the capabilities of enterprises in the field in which they operate and use Industry 4.0 maturity models in the literature. It was also found that the dimensions used in these models are mostly similar to each other.

Future research areas can be carried out with larger samples to measure the Industry 4.0 maturity levels of companies from different sectors and to determine the importance levels of evaluation criteria. The models can be adapted to the sectors by making changes to existing models.

This study aims to measure the size-based and overall maturity levels of enterprises and to identify the obstacles to the transition of enterprises to Industry 4.0. The majority of the 61 companies tested under the maturity model in our study still appear immature or partially mature. There is a need for a lot of improvements and a reassessment of transformation strategies related to Industry 4.0. The findings also suggest that companies should continue to strive for transformation.

The government should force all manufacturing industries to adopt Industry 4.0, as transformation is inevitable. The lack of national policies can be a problem as Industry 4.0 is a technological revolution. Manufacturing industries should be aware of global developments to seize the opportunity to compete. To fully realize the potential of Industry 4.0 and reduce the risk of loss of investment, the enterprise must take into account all possible difficulties in all aspects during strategic planning.

The findings show that the application of Industry 4.0 differs according to the sector and the type of business. We think this will represent a promising avenue for future research. It is strongly recommended to create a roadmap for Industry 4.0 integration for enterprises based on our findings. These findings can serve as guidelines for practitioners and researchers.

This study sheds light on the search for a suitable maturity model that will support companies' work on Industry 4.0 in a competitive environment. Various maturity models are proposed in the literature; however, six dimensions have become fundamental for companies that evaluate new opportunities and adopt transformation solutions. Our findings show that our maturity model is suitable for conducting self-assessment in manufacturing enterprises. Awareness seminars can be organized by non-governmental organizations to promote the use of Industry-4.0-related technologies. Especially in face-to-face surveys, it has been observed that some businesses are not very aware of the benefits of Industry 4.0 technologies. Some businesses have different concerns, such as that their investments will

not return. The government can hold informational meetings on their incentive practices for these enterprises.

Finally, the results provide valuable contributions to the field of maturity models for Industry 4.0, both for interested researchers and professionals embarking on the I4.0 journey.

**LIMITATIONS:** The SANOL Maturity Model was applied to corporations established in a limited area, namely, at the Ankara Chamber of Industry Organized Park in Turkey. The survey was applied to a total of 87 businesses. Twenty-six were traditional factories and stated that they were not interested in Industry 4.0. Sixty-one factories were in the process of transition to digitalization.

The model could also be applied in other industrial parks in Turkey and different countries. The impact of geographical differences on maturity level could be investigated. The number of model dimensions and questions could be increased in future studies.

The level of interest and lack of knowledge across the participants were among the limitations of the study.

The lack of standards for the readiness and maturity models was also an important limitation.

**Supplementary Materials:** The following supporting information can be downloaded at: https://www.mdpi.com/article/10.3390/su14159478/s1, Industry 4.0 Maturity Index.

**Author Contributions:** Conceptualization, H.Y. and C.S.; methodology, C.Ü.; formal analysis, C.Ü. and H.Y.; resources, C.S.; writing—original draft preparation, C.Ü. and H.Y.; writing—review and editing, C.Ü., H.Y. and C.S.; visualization, C.Ü. and H.Y.; supervision, C.S. and C.Ü.; project administration, H.Y. All authors have read and agreed to the published version of the manuscript.

**Funding:** This research received no external funding.

**Institutional Review Board Statement:** Not applicable.

**Informed Consent Statement:** Informed consent was obtained from all subjects involved in the study.

**Data Availability Statement:** Not applicable.

**Acknowledgments:** Authors of the paper are grateful to the editors and reviewers for their invaluable contribution.

**Conflicts of Interest:** Authors of the paper declare no conflict of interest.

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
