# Peer review of "Application of the Maturity Model in Industrial Corporations"

_sustainability, doi:10.3390/su14159478_

Round 1

Reviewer 1 Report

                                                                                                                                  30.5.2022

Ref: Sustainability Manuscript ID :  Sustainability 1749070

Name of the article: Application of SANOL Maturity Model in Industrial Corporations at Ankara Chamber of Industry Industrial Park

General

The article analyses the companies with features described by the Industry 4.0 paper of the year 2011. The analysis is made with AHP model called SANOL. The companies are located in an industrial park in Ankara,  Turkey. The authors say that they have added to the maturity model the dimensions regularization and incentives. This would be positive for the sake of the possibility to apply the model in decision making.

The topic is a most interesting and also actual. The industrial countries have been competing strongly in industries in the era of globalization. As the developing countries have entered into the field of traditional industry fields with low-cost labor and cheap raw materials, it is natural the companies of industrialized countries are trying to expand to the fields where the more advanced production factors like education, RTI (Research, Technology and Innovation) and other specialized skills are employed. 

The industry 4.0. was very ambitious program even in Germany where it was published in the year 2011. It is understandable that industrial base of start-up companies in Turkey is thin. And to make research in Ankara can be difficult in this field. This feature is reflecting in the whole article draft and case review. Some of the mother companies can be large. The objects of the study are smaller “departments” or daughter companies to the large corporations.

The objective in this article is ton describe the construction of Industry 4.0. maturity level model. It is based on expert opinions and AHP. Validity test is made according to the authors in the field with expert interviewees. The focus in the paper is maturity and readiness. The paper encourages the companies to improve their maturity and readiness for adopting.

The paper sets the industry 4.0 to perspective by describing the industry and society changes through industrial revolutions. The main cause for industry 4.0 revolution is the technology change to digitalization, of which we have seen only a small part. It seems that in Turkey the strategy in Turkey the strategy for roadmap for the industry 4.0 is mainly adopting the new situation parameters.

The strengths of the article

The strong part in the article is the list and explanation of main dimensions, and technologies relating to the evolution of industry 4.0. These dimensions are connected to the number of new technologies relating to industry 4.0. The list of the technologies is the base of the industrial survey. The number of dimensions in the study varied between three and thirteen. Another strong point in the article is the literature review in the field.

The method of the article

The study uses AHP method to level modeling. The main dimensions (together six pieces) are determined by the weights. Maturity levels are categorized to final levels. According to the analysis the category levels are determined by the weight numbers. Then accordingly the maturity index is formed. As a non-expert in this method I can write that all what I see looks sensible. I let, however, the AHP experts to evaluate eventually the soundness of the use of the method.

The results of the study

The part of the results which is clear are the weights of the main dimensions: strategy and management, technology, and data and security. These results are clear enough, significant and understandable.

Commenting the other results is difficult. They do not assure me. I believe due to the background of the developing stage of the Turkish industry and the stat-up companies the answers are more or less centered to the middle of the scale, so the are 3 +- (plus minus) 0.5., except some outliers in the answers. This mean that they are indifferent. It is difficult to make any relevant conclusions out of them. I do not know whether it is possible to make some kind of statistical significance analysis with AHP method. This could possibly disclose, whether the results are meaning something or not. I have a strong doubt on them.

Weakness of the article

The validity pf the results is doubtful. This should be improved and shown explicitly – in one way or another.

Recommendations and proposals to improve the article

The articled should be made clearer in many ways:

1.      Describe the companies more closely. Try to show the technologies they are using and the ones they stride to improve applying something new. Use concrete examples.

2.       Try to describe the technologies more thoroughly and the steps made toward expanded application of them more widely. Naming them alone is not enough, you should make the applications concrete.  – Here maybe narrower view is deeper and better than wider view.

3.       The differences in numbers should be made clearer to be assuring. Small differences are a weak base for conclusions.

4.       You should think about your tittle in the article. In any case you should delete the references to Ankara and Turkey. They should not appear in the article tittle. I should propose to rename the article so that the name includes terms Industry 4.0 and maturity. You have to invent the name which is better describing the content. You do not need the term SANOL in the tittle nor the abstract. It is the result of modeling.

Author Response

Thank you for your suggestions and contribution. The responses to Reviewer #1 comments are as follows. English grammar errors have been corrected by a language expert.

Reviewer 2 Report

The abstract should be structured in a single paragraph.

The content of table 1 can be summarized in the text. Since it is not a core information of the research, it should not be presented in a table. In the introduction, it is recommended to avoid tables.

The introduction should focus on presenting the relevance of the study theme and the gap in the literature. It should end presenting the research goals and the next sections. There is several information in the introduction that should be in the methods section. The relevance of the study and the gap in the literature must be further explored (there are only two paragraphs and 2 lines for this now and it is not presenting the necessary information).

The section of theoretical background is expected to present the current literature on the theme. Section 2.1 currently presents definitions for some technologies of Industry 4.0. Definitions can be presented, but they are not enough for the theoretical background.

The authors should review the column of disadvantages presented in the table 2. Stating that “The number of dimensions is inadequate for a plausible analysis”, for example, seems a subjective argue. The models you are analysing were published in high quality journals. To make criticisms, you must present a robust and objective analysis. In addition, for the purpose of a theoretical background, authors could provide further information and analysis on the models evaluated.

To make clearer the steps of the model development and the research as a whole, I recommend the authors to add a figure presenting the steps conducted.

The model presented 6 dimensions and 22 sub-dimensions, why the questionnaire presented 58 questions? What were these questions? This information should be clear in the text. Maybe add the questionnaire in the appendix.

Authors should review the structure of the methods section. The first paragraph summarizes procedures that are detailed later, but it makes more difficult to understand the logical flow of the research. Authors should explain it chronologically to make it clearer.

Also in the methods section, there are procedures and equations that do not have the reference used for it. It is important to add it.

Shouldn’t the dimension weight be in the results section? In addition, authors should show all the steps to achieve the weights distribution.

How do the authors developed Table 5? It is not clear in the text. Also, make it clear if it was used in the survey with the companies.

Before presenting the data analysis, authors should introduce the sample of companies.  

The debates with the literature are missing after results presentation. 

Author Response

Thank you for your suggestions and contribution. The responses to Reviewer #2 comments are as follows. English grammar errors have been corrected by a language expert.

Reviewer 3 Report

The manuscript contributes an industry 4.0 maturity model. The content is actual and interesting. However, it contains some critical opportunities for improvement. Some of the most critical issues are:

-     -    the writing style and microstructure of the content

-       - the methodological description (research design) of the model is mixed with the research report,

-      -  weak analysis of the previous models (disadvantages)

-       - unclear determination of population and the units of observation,

-       -  questionable use of Incentives and Regulations dimensions (the problem of the units of observation)

-      -  weak or no connection of the dimensionality of the proposed model to the previous models

-        - missing scientific support(citation of previous research) for the methods of assessment of the weights and the calculation of the index,

-        - missing inference statistics

-      -   weak or no correspondence between introduction (objectives, hypotheses) and conclusion,

-       -  missing references to other research results in conclusion,

-        - justification of the importance of the research only referencing Turkish research.

The manuscript deals with an important topic that could contribute to the scientific field and support policy and companies' decision-making. I recommend rewriting the manuscript considering the rules of scientific writing. Moreover, the methodological appraoch should be reconsidered on the ground of appropriate previous research.

Author Response

Thank you for your suggestions and contribution. The responses to Reviewer #3 comments are as follows. English grammar errors have been corrected by a language expert.

Reviewer 4 Report

Dear Authors, sincere complements on a very good concept. The area of digital maturity is interesting and some hard work seems to of been done. I do like the overall concept and approach but i feel the key oppertunity for improvement are around the academic writing of the paper.

Major correction: Please rewrite and restructure the paper to be more academic, from the structure of the abstract and through the paper. A strong example is the methods section which needs significant literature reinforcing with a clear definition of all methods. 

The results must provide for details of the maturity matrix. We should see much more details of the companies that the model was deployed at. 

Statistically is the sample sufficient? please provide us with a statistical calculation to prove.

The data collection approach, initial analysis and statistical analiys that verifies internal consitency etc is not provided. 

The paper can be significantly improved in the unpacking of the results in a structured manner.

Stronger justification of the applicability of the model would also assist.

wishing you all of the very best

Author Response

Thank you for your suggestions and contribution. The responses to Reviewer #4 comments are as follows. English grammar errors have been corrected by a language expert.

Round 2

Reviewer 1 Report

The authors have improved the clarity of the paper. The methodological part is now very thorough, and therefore a bit long. But now it is detailed, and maybe somebody may repeat this study in some other country using the method as a model. 

I think the paper is ready to be published.

Author Response

Thank you for your suggestions and contribution.

Reviewer 2 Report

Several important points indicated in the previous revision were not addressed:

The introduction was not improved as required. The relevance and gap in the literature considering articles of international journals must be further explored. The introduction text still does not present the context, relevance and gap.

This comment is still a problem that was not solved: The section of theoretical background is expected to present the current literature on the theme. Section 2.1 currently presents definitions for some technologies of Industry 4.0. Definitions can be presented, but they are not enough for the theoretical background.

This comment is still a problem that was not solved: The authors should review the column of disadvantages presented in the table 2. Stating that “The number of dimensions is inadequate for a plausible analysis”, for example, seems a subjective argue. The models you are analysing were published in high quality journals. To make criticisms, you must present a robust and objective analysis. In addition, for the purpose of a theoretical background, authors could provide further information and analysis on the models evaluated. – the authors should present objective analysis of the models.

This comment is still a problem that was not solved: To make clearer the steps of the model development and the research as a whole, I recommend the authors to add a figure presenting the steps conducted.

The lines 314-315 do not explain this difference in the number of questions: The model presented 6 dimensions and 22 sub-dimensions, why the questionnaire presented 58 questions? What were these questions? This information should be clear in the text.

This comment is still a problem that was not solved: Authors should review the structure of the methods section. The first paragraph summarizes procedures that are detailed later, but it makes more difficult to understand the logical flow of the research. Authors should explain it chronologically to make it clearer.

This comment is still a problem that was not solved: Shouldn’t the dimension weight be in the results section? In addition, authors should show all the steps to achieve the weights distribution.

This comment is still a problem that was not solved: The debates with the literature are missing after results presentation. 

Author Response

(The authors gave the same response as above.)

Reviewer 3 Report

The manuscript presents sound research on the maturity of enterprises regarding Industry 4.0. It adds an interesting piece of knowledge to the scientific field. However, it is pretty poorly written.

Hence it needs substantial improvement. The text should be restructured, many passuses should be rewritten. Besides improving the quality of writing, the manuscript calls for restructuring of the reporting, where the presentation of the maturity model and the report on results of the specific application of the model should be given separately.

Author Response

(The authors gave the same response as above.)

Reviewer 4 Report

Thank you for the updates

unfortunately the fundamental problems highlighted have not been adressed.

Author Response

(The authors gave the same response as above.)

Round 3

Reviewer 2 Report

Most of the comments were addressed in this second revision. Just three minor but important adjustments are presented:

The authors improved the introduction this time, however, three paragraphs were added without any reference. In scientific research, the information that does not result from the study’s findings must be based on the literature. The information presented in these paragraphs is not originally from the authors and should be based on references.

The information mentioned in the Lines 738-750 should also be mentioned in the introduction.

The title of the section 2.2 could be more specific, focusing on the core information presented in the section. 

Author Response

(The authors gave the same response as above.)

Reviewer 3 Report

Well done. I have only now seen the problem with the Incentive and regulations dimension. The name does not express its meaning. I suggest considering renaming the dimension.  

Besides, the dimension remains unclearly defined. Moreover, some more confusion is introduced with the statement:

"Model we developed consists of Strategy and Management, Customers and Suppliers, Employees and Corporate Culture, Technology, Data and Security, Legal Regulations, and Benefiting from Incentives."

where instead of Regulation and incentives, we have two concepts.

Nevertheless, the definition of the newly introduced dimension should be improved. 

The writing style is still at a relatively low level. I suggest detailed proofreading of the manuscript..

Author Response

(The authors gave the same response as above.)
